# Local and Global Modeling with Large Language Models for Time Series Forecasting

## Abstract

Time series forecasting is critical across multiple domains, where time series data exhibits both local patterns and global dependencies. While Transformer-based methods effectively capture global dependencies, they often overlook short-term local variations in time series. Recent methods that adapt large language models (LLMs) into time series forecasting inherit this limitation by treating LLMs as black-box encoders, relying solely on the final-layer output and underutilizing hierarchical representations. To address this limitation, we propose Logo-LLM, a novel LLM-based framework that explicitly extracts and models multi-scale temporal features from different layers of a pre-trained LLM. Through empirical analysis, we show that shallow layers of LLMs capture local dynamics in time series, while deeper layers encode global trends. Moreover, Logo-LLM introduces lightweight Local-Mixer and Global-Mixer modules to align and integrate features with the temporal input across layers. Extensive experiments demonstrate that Logo-LLM achieves superior performance across diverse benchmarks, with strong generalization in few-shot and zero-shot settings while maintaining low computational overhead.

## 1 Introduction

Time series forecasting is a critical task across various domains, where temporal data naturally exhibit multi-scale patterns, including short-term local variations and long-range global dependencies. Local features capture transient behaviors within short time windows, while global dependencies reflect correlations across broader temporal spans. Early Transformer-based models (Liu et al., 2022b; Zhou et al., 2022b; Wu et al., 2021; Liu et al., 2021) have emerged as the dominant approach due to their remarkable capacity to model global dependencies via the attention mechanism. However, these methods often neglect local temporal patterns, limiting their ability to capture fine-grained fluctuations. To address this, Nie et al. proposes a patch-based framework that significantly models local variations, emphasizing short-term dynamics in time series data. This perspective has spurred a series of subsequent studies (Chen et al., 2024; Zhang et al., 2024) aiming to explicitly capture local and global temporal patterns in a multi-scale manner.

Despite these advancements, recent efforts to adapt large language models (LLMs) for time series (Liu et al., 2025; Jin et al., 2024; Zhou et al., 2023) appear to inherit the same limitations. While LLMs inherit the global modeling strengths of Transformer, existing approaches typically treat the LLM as a black-box encoder, leveraging only the final-layer output for prediction. This practice underutilizes the rich hierarchical representations distributed across different layers, resulting in a persistent bias towards global patterns and a neglect of local temporal variations. To apply the LLM more effectively, we first analyze the internal representational properties of LLMs in the context of time series. As shown in Figure 1, our empirical analysis reveals a clear layer-wise specialization: shallow layers are more sensitive to local dynamics, while deeper layers encode broader temporal dependencies. This insight motivates us to design a new paradigm that explicitly extracts and integrates multi-layer LLM features, leveraging shallow-layer outputs as local representations and deep-layer outputs as global representations.

Inspired by the above insights, we propose a **Lo**cal and **glo**bal modeling **LLM**-based framework called **Logo-LLM** that models local and global features through LLM's representations. Logo-LLM employs a pre-trained GPT2 (Radford et al., 2019) backbone to extract multi-layer semantic

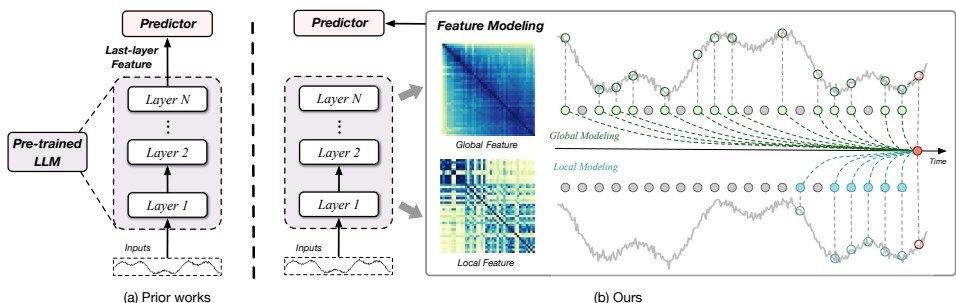

(a) Prior works                    (b) Ours

Figure 1: Comparison of LLM usage paradigms. Prior works treat LLMs as black-box encoders and use only the last-layer feature. Our method explicitly extracts features from multiple layers, leveraging shallow-layer features for local modeling and deep-layer features for global modeling, enabling a more fine-grained understanding of temporal dynamics.

representations, explicitly using the early-layer features to capture local temporal variations and the deep-layer features to encode global dependencies. To effectively align these heterogeneous features with temporal data, we design two specialized modules: the Global-Mixer, which integrates global representations to model long-range dependencies, and the Local-Mixer, which aligns local representations to capture short-term fluctuations. Decoupling local and global modeling enables more precise capture of multi-scale temporal patterns, addressing the limitation of relying solely on final-layer outputs. The experimental results on real-world benchmarks demonstrate that Logo-LLM consistently achieves superior performance in long-term forecasting, with strong generalization in few-shot and zero-shot settings under limited data. Notably, our approach maintains competitive complexity, offering an efficient yet powerful alternative for temporal modeling with LLMs.

Here we summarize our key contributions as follows:

1. We propose a Local and global modeling LLM-based framework (Logo-LLM) to achieve superior performance in time series forecasting tasks supported by extensive experiments, including long-term, few-shot, and zero-shot scenarios. To the best of our knowledge, this study is the first to propose leveraging the internal layer-wise semantics of a pre-trained LLM to separately capture local and global variations in time series data.

2. We design Local-Mixer and Global-Mixer modules to separately align local and global features with the temporal data and capture long- and short-term variations in time series. Decoupling local and global features allows the model to better exploit the hierarchical nature of LLMs rather than rely solely on the final-layer output.

3. Extensive experiments on multiple real-world benchmark datasets demonstrate that Logo-LLM achieves superior performance in long-term forecasting, few-shot, and zero-shot learning tasks, with favorable generalization ability and low computational complexity.

## 2  RELATED WORK

### 2.1  LLMS FOR TIME SERIES

With the great popularity of LLMs in the NLP field, the application of LLMs into time series tasks has emerged as an innovative and promising approach. LLM-based methods leverage the extensive world knowledge acquired through pre-training to enhance contextual modeling capabilities in time series analysis. Zhou et al. demonstrated the potential of fine-tuning and adapting LLMs for time series forecasting. Jin et al. reprogrammed time series inputs by text-based prototypes and augmented them by treating the context as a prefix to achieve improved alignment with LLMs. Liu et al. proposed a framework that incorporates specific modules to align textual data with temporal data. Despite these advances, these methods ignore the intrinsic local features of temporal data, resulting in the full potential of LLMs' general knowledge being underutilized.

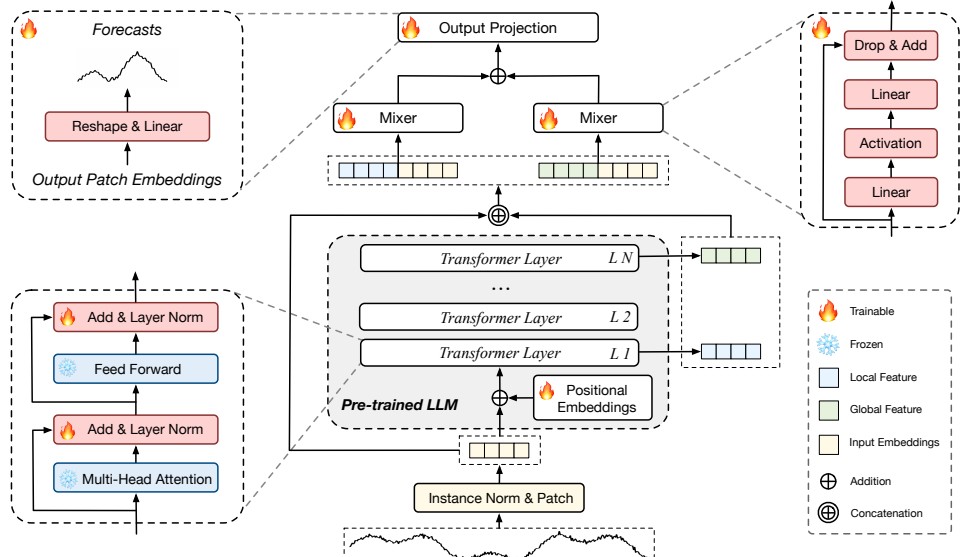

Figure 2: Overview of the proposed Logo-LLM framework. Logo-LLM extracts intermediate representations from multiple layers of a pre-trained LLM to explicitly model local and global temporal patterns. Two specialized Mixer modules are introduced to align these hierarchical features with the temporal input, enabling fine-grained modeling of local and global variations. Most LLM parameters are kept frozen, enabling efficient adaptation under limited supervision.

## 2.2 LOCAL AND GLOBAL MODELING FOR TIME SERIES

Time series data inherently exhibit both local and global variations. Transformer-based architectures (Liu et al., 2022b; Zhou et al., 2022b; Wu et al., 2021; Nie et al., 2023) have demonstrated superior ability in capturing global patterns. However, standard Transformers suffer from high computational costs for long sequences and may overlook crucial local details. To address this, Wang et al. adopts a multi-scale isometric convolutional network to separately capture local and global contexts. In the CV and NLP domain, recent studies (Li et al., 2022; Chen et al., 2023; Lee et al., 2024; Liu et al., 2024) observed that Transformer models naturally learn fine-grained local textures in shallow layers, while deep layers capture high-level semantics and global dependencies. Motivated by this insight, we introduce a hierarchical feature modeling paradigm, progressing from local to global representations, into time series forecasting.

## 3 METHODOLOGY

Our model architecture is illustrated in Figure 2. Given a time series input $\mathbf{X} \in \mathbb{R}^{L \times D}$ of length $L$, our goal is to leverage the multi-layer features generated by a pre-trained LLM. Specifically, the input is first processed through instance normalization and patching, and then passed into the LLM to obtain hierarchical features. Unlike previous works that treat LLMs as black-box encoders using only final-layer features, we decouple temporal modeling by explicitly leveraging shallow-layer and deep-layer outputs. These features are concatenated with the patching input and fed into corresponding Mixer modules, which serve to align the input with the learned temporal representations and enhance the modeling of temporal dynamics. Finally, a fusion step integrates the aligned features by an output projection layer.

**Input Transformations.** Given a multivariate time series $\mathbf{X} \in \mathbb{R}^{L \times D}$, we split it into $D$ univariate time series $\mathbf{X}_i \in \mathbb{R}^{L \times 1}$ along the channel dimension and process them independently, following the Channel-Independence strategy proposed in Zeng et al.. To ensure a similar distribution, we apply

instance normalization (Kim et al., 2021) to each univariate series. The normalization is defined as:

$$\mu_i = \frac{1}{L} \sum_{t=1}^{L} \mathbf{X}_{i,t}, \quad \sigma_i = \frac{1}{L} \sum_{t=1}^{L} (\mathbf{X}_{i,t} - \mu_i), \quad \bar{\mathbf{X}}_{i,t} = \frac{\mathbf{X}_{i,t} - \mu_i}{\sqrt{\sigma_i + \epsilon}}, \tag{1}$$

where $\mu_i, \sigma_i$ denote the mean and variance of the $t$-th time step of the $i$-th univariate series and $\epsilon$ is a small constant for numerical stability. Inspired by Nie et al., we further segment each normalized series $\bar{\mathbf{X}}_i$ into overlapping patches of length $P$ and stride $S$. The technique can allow the model to see the longer contextual sequence, which can significantly improve prediction performance.

$$\bar{\mathbf{X}}_{i,j} = \text{Unfold}(\text{ReplicationPad}(\bar{\mathbf{X}}_i), \text{size} = P, \text{step} = S), \tag{2}$$

where $\bar{\mathbf{X}}_{i,j}$ is the $j$-th patch of the $i$-th input series.

**Pre-trained LLM Backbone.** We design a lightweight input embedding layer to adapt raw temporal data for integration with the pre-trained LLM. The embedding layer linearly projects each patch into the token space, formulated as $\tilde{\mathbf{X}}_i = \text{TE}(\bar{\mathbf{X}}_i)$, where $\text{TE}(\cdot)$ denotes a learnable token embedding. To preserve temporal ordering, we also incorporate a positional embedding layer $\text{PE}(\cdot)$. The embedded inputs are then fed into $N$ Transformer blocks of a pre-trained LLM, with only $\text{PE}(\cdot)$ and layer normalization layers fine-tuned to significantly reduce the number of trainable parameters. Notably, self-attention and FFN layers remain untouched to retain the rich knowledge obtained through pre-training. The process is described as:

$$\begin{aligned} \bar{\mathbf{H}}_i &= \tilde{\mathbf{X}}_i + \text{PE}(\bar{\mathbf{X}}_i), \\ \bar{\mathbf{H}}_i^{(n)} &= \mathcal{T}^{(n)}(\bar{\mathbf{H}}_i^{(n-1)}), \quad n \in \{1, \cdots, N\}, \end{aligned} \tag{3}$$

where $\mathcal{T}^{(n)}, \bar{\mathbf{H}}_i^{(n)}$ are the $n$-th Transformer block and corresponding hidden state of the $i$-th patch.

**Mixer Modules.** For each patch, the pre-trained LLM captures temporal dependencies through stacked Transformer layers, progressively encoding local and global patterns. Inspired by hierarchical representation learning in vision models, we leverage intermediate LLM outputs to disentangle temporal features at different levels of abstraction. Specifically, we extract shallow-layer feature $\bar{\mathbf{H}}_i^{(0)}$ to represent local short-term variations and deep-layer feature $\bar{\mathbf{H}}_i^{(N)}$ to capture broader temporal trends. This design provides a clearer semantic separation between local and global temporal patterns. As illustrated in Figure 1, shallow-layer outputs preserve high-resolution temporal details, while deeper layers abstract long-range patterns, justifying our decoupled design. Moreover, the design improves computational efficiency by avoiding aggregation across multiple layers.

Furthermore, we employ two lightweight MLP-based Mixer modules (Local-Mixer and Global-Mixer) to align these hierarchical features with the input patches. These modules enhance non-linearity and fuse semantic representations from the LLM with temporal sequences, facilitating more accurate forecasting. The fused results are then aggregated through element-wise addition and projected via a linear layer to produce the final predictions.

$$\begin{aligned} \bar{\mathbf{H}}_{i,l} &= \tilde{\mathbf{X}}_i + \text{Dropout}\left(\mathbf{W}_l^{(2)} \cdot \phi\left(\mathbf{W}_l^{(1)} \cdot [\tilde{\mathbf{X}}_i \,\|\, \bar{\mathbf{H}}_i^{(0)}]\right)\right), \\ \bar{\mathbf{H}}_{i,g} &= \tilde{\mathbf{X}}_i + \text{Dropout}\left(\mathbf{W}_g^{(2)} \cdot \phi\left(\mathbf{W}_g^{(1)} \cdot [\tilde{\mathbf{X}}_i \,\|\, \bar{\mathbf{H}}_i^{(N)}]\right)\right), \\ \mathbf{Y}_i &= \mathbf{W}_{out} \cdot \text{Reshape}\left(\bar{\mathbf{H}}_{i,l} + \bar{\mathbf{H}}_{i,g}\right), \end{aligned} \tag{4}$$

where $[: \,\|\, :]$ and $\phi$ denote $\text{Concat}(\cdot)$ and $\text{ReLU}(\cdot)$ functions. $\mathbf{W}_l^{(1)}, \mathbf{W}_l^{(2)}$ and $\mathbf{W}_g^{(1)}, \mathbf{W}_g^{(2)}$ are the MLP weights in the Local-Mixer and Global-Mixer modules, respectively. $\mathbf{W}_{out}$ denotes the weight matrix of output projection layer. $\mathbf{Y}_i$ is the final results of the $i$-th patch.

## 4 EXPERIMENTAL RESULTS

**Experimental Settings.** In various benchmarks, Logo-LLM consistently outperforms state-of-the-art time series forecasting methods, especially in few-shot and zero-shot learning. We compare our approach with a wide range of recent models, especially fine-tuning time series analysis language models, including CALF Liu et al. (2025), Time-LLM Jin et al. (2024), and GPT4TS Zhou et al.

Table 1: Long-term time series forecasting. Best results are in **bold** and second are in underlined.

| Methods | Logo-LLM (Ours) | | CALF (2025) | | Time-LLM (2024) | | GPT4TS (2023) | | iTransformer (2023a) | | PatchTST (2023) | | TimesNet (2023) | | FEDformer (2022b) | | DLinear (2023) | |
|---|---|---|---|---|---|---|---|---|---|---|---|---|---|---|---|---|---|---|
| Metric | MSE | MAE | MSE | MAE | MSE | MAE | MSE | MAE | MSE | MAE | MSE | MAE | MSE | MAE | MSE | MAE | MSE | MAE |
| ETTm1 96 | **0.317** | **0.343** | 0.323 | 0.350 | 0.359 | 0.381 | 0.329 | 0.364 | 0.341 | 0.376 | 0.328 | 0.367 | 0.338 | 0.375 | 0.379 | 0.419 | 0.345 | 0.372 |
| ETTm1 192 | **0.368** | **0.370** | 0.374 | 0.375 | 0.383 | 0.393 | **0.368** | 0.382 | 0.382 | 0.395 | **0.368** | 0.385 | 0.374 | 0.387 | 0.426 | 0.441 | 0.380 | 0.389 |
| ETTm1 336 | **0.394** | **0.393** | 0.409 | 0.399 | 0.416 | 0.414 | 0.400 | 0.403 | 0.418 | 0.418 | 0.399 | 0.410 | 0.410 | 0.411 | 0.445 | 0.459 | 0.413 | 0.413 |
| ETTm1 720 | **0.460** | **0.430** | 0.477 | 0.438 | 0.483 | 0.449 | 0.460 | 0.439 | 0.487 | 0.456 | 0.454 | 0.439 | 0.478 | 0.450 | 0.543 | 0.490 | 0.474 | 0.453 |
| ETTm2 96 | **0.175** | **0.253** | 0.178 | 0.256 | 0.193 | 0.280 | 0.178 | 0.263 | 0.185 | 0.272 | 0.183 | 0.270 | 0.187 | 0.267 | 0.203 | 0.287 | 0.193 | 0.292 |
| ETTm2 192 | **0.243** | **0.298** | 0.245 | 0.300 | 0.257 | 0.318 | 0.245 | 0.306 | 0.253 | 0.313 | 0.255 | 0.314 | 0.533 | 0.563 | 0.269 | 0.328 | 0.284 | 0.362 |
| ETTm2 336 | **0.304** | **0.338** | 0.309 | 0.341 | 0.317 | 0.353 | 0.309 | 0.347 | 0.315 | 0.350 | 0.309 | 0.347 | 0.321 | 0.351 | 0.325 | 0.366 | 0.369 | 0.427 |
| ETTm2 720 | 0.408 | 0.400 | **0.402** | **0.395** | 0.419 | 0.411 | 0.409 | 0.408 | 0.413 | 0.406 | 0.412 | 0.404 | 0.408 | 0.403 | 0.421 | 0.415 | 0.554 | 0.522 |
| ETTh1 96 | 0.373 | **0.388** | **0.369** | 0.389 | 0.398 | 0.410 | 0.376 | 0.397 | 0.386 | 0.404 | 0.393 | 0.408 | 0.384 | 0.402 | 0.376 | 0.419 | 0.386 | 0.400 |
| ETTh1 192 | **0.413** | **0.412** | 0.427 | 0.423 | 0.451 | 0.440 | 0.438 | 0.426 | 0.441 | 0.436 | 0.445 | 0.434 | 1.008 | 0.792 | 0.420 | 0.448 | 0.437 | 0.432 |
| ETTh1 336 | **0.434** | **0.424** | 0.456 | 0.436 | 0.508 | 0.471 | 0.479 | 0.446 | 0.489 | 0.461 | 0.484 | 0.451 | 0.491 | 0.469 | 0.459 | 0.465 | 0.481 | 0.459 |
| ETTh1 720 | **0.446** | **0.447** | 0.479 | 0.467 | 0.483 | 0.478 | 0.495 | 0.476 | 0.508 | 0.493 | 0.480 | 0.471 | 0.521 | 0.500 | 0.506 | 0.507 | 0.519 | 0.516 |
| ETTh2 96 | **0.274** | **0.330** | 0.284 | 0.336 | 0.295 | 0.346 | 0.295 | 0.348 | 0.300 | 0.349 | 0.294 | 0.343 | 0.340 | 0.374 | 0.358 | 0.397 | 0.333 | 0.387 |
| ETTh2 192 | **0.344** | **0.376** | 0.353 | 0.380 | 0.386 | 0.399 | 0.386 | 0.404 | 0.379 | 0.398 | 0.377 | 0.393 | 0.402 | 0.414 | 0.429 | 0.439 | 0.477 | 0.476 |
| ETTh2 336 | 0.369 | 0.398 | **0.362** | **0.394** | 0.447 | 0.443 | 0.421 | 0.435 | 0.418 | 0.429 | 0.381 | 0.409 | 0.452 | 0.452 | 0.496 | 0.487 | 0.594 | 0.541 |
| ETTh2 720 | **0.394** | **0.417** | 0.406 | 0.428 | 0.428 | 0.444 | 0.422 | 0.445 | 0.428 | 0.445 | 0.412 | 0.433 | 0.462 | 0.468 | 0.463 | 0.474 | 0.831 | 0.657 |
| ILI 24 | 1.683 | **0.800** | 1.672 | 0.841 | **1.651** | 0.841 | 1.869 | 0.823 | 2.321 | 0.937 | 2.221 | 0.883 | 1.826 | 0.893 | 2.721 | 1.133 | 5.060 | 1.709 |
| ILI 36 | 1.825 | **0.816** | 1.725 | 0.872 | **1.701** | 0.861 | 1.853 | 0.854 | 2.188 | 0.945 | 2.313 | 0.904 | 2.678 | 0.986 | 2.768 | 1.118 | 4.413 | 1.549 |
| ILI 48 | **1.777** | **0.793** | 1.937 | 0.937 | 2.153 | 1.041 | 1.886 | 0.855 | 2.231 | 0.956 | 2.048 | 0.886 | 2.584 | 0.937 | 2.637 | 1.088 | 4.109 | 1.473 |
| ILI 60 | **1.748** | **0.807** | 2.128 | 0.999 | 2.064 | 0.953 | 1.877 | 0.877 | 2.292 | 0.991 | 2.008 | 0.915 | 1.980 | 0.894 | 2.696 | 1.050 | 4.233 | 1.481 |
| Weather 96 | 0.173 | **0.206** | **0.168** | 0.207 | 0.195 | 0.233 | 0.182 | 0.223 | 0.174 | 0.214 | 0.177 | 0.218 | 0.172 | 0.220 | 0.217 | 0.296 | 0.196 | 0.255 |
| Weather 192 | 0.217 | **0.247** | **0.216** | 0.251 | 0.240 | 0.269 | 0.231 | 0.263 | 0.221 | 0.254 | 0.225 | 0.259 | 0.219 | 0.261 | 0.276 | 0.336 | 0.237 | 0.296 |
| Weather 336 | 0.275 | **0.289** | **0.271** | 0.292 | 0.293 | 0.306 | 0.283 | 0.300 | 0.278 | 0.296 | 0.278 | 0.297 | 0.280 | 0.306 | 0.339 | 0.380 | 0.283 | 0.335 |
| Weather 720 | **0.350** | **0.339** | 0.355 | 0.352 | 0.368 | 0.354 | 0.360 | 0.350 | 0.358 | 0.349 | 0.354 | 0.348 | 0.365 | 0.359 | 0.403 | 0.428 | 0.345 | 0.381 |
| ECL 96 | 0.167 | 0.246 | **0.147** | **0.240** | 0.204 | 0.293 | 0.185 | 0.272 | 0.148 | **0.240** | 0.195 | 0.285 | 0.168 | 0.272 | 0.193 | 0.308 | 0.197 | 0.282 |
| ECL 192 | 0.176 | 0.254 | 0.163 | 0.254 | 0.207 | 0.295 | 0.189 | 0.276 | **0.162** | **0.253** | 0.199 | 0.289 | 0.184 | 0.289 | 0.201 | 0.315 | 0.196 | 0.285 |
| ECL 336 | 0.190 | 0.270 | **0.178** | 0.270 | 0.219 | 0.308 | 0.204 | 0.291 | **0.178** | 0.269 | 0.215 | 0.305 | 0.198 | 0.306 | 0.214 | 0.329 | 0.209 | 0.301 |
| ECL 720 | 0.228 | 0.303 | **0.215** | 0.300 | 0.263 | 0.341 | 0.245 | 0.324 | 0.225 | 0.317 | 0.256 | 0.337 | 0.220 | 0.320 | 0.246 | 0.355 | 0.245 | 0.333 |
| Traffic 96 | 0.434 | **0.252** | 0.416 | 0.274 | 0.536 | 0.359 | 0.468 | 0.307 | **0.395** | 0.268 | 0.544 | 0.359 | 0.593 | 0.321 | 0.587 | 0.366 | 0.650 | 0.396 |
| Traffic 192 | 0.450 | **0.257** | 0.430 | 0.276 | 0.530 | 0.354 | 0.476 | 0.311 | **0.417** | 0.276 | 0.540 | 0.354 | 0.617 | 0.336 | 0.604 | 0.373 | 0.598 | 0.370 |
| Traffic 336 | 0.466 | **0.264** | 0.451 | 0.286 | 0.530 | 0.349 | 0.488 | 0.317 | **0.433** | 0.283 | 0.551 | 0.358 | 0.629 | 0.336 | 0.621 | 0.383 | 0.605 | 0.373 |
| Traffic 720 | 0.500 | **0.283** | 0.478 | 0.301 | 0.569 | 0.371 | 0.521 | 0.333 | **0.467** | 0.302 | 0.586 | 0.375 | 0.640 | 0.350 | 0.626 | 0.382 | 0.645 | 0.394 |
| Exchange 96 | **0.081** | **0.198** | 0.083 | 0.203 | 0.123 | 0.251 | 0.096 | 0.218 | 0.086 | 0.206 | 0.088 | 0.205 | 0.107 | 0.234 | 0.148 | 0.278 | 0.088 | 0.218 |
| Exchange 192 | **0.176** | **0.295** | 0.186 | 0.306 | 0.224 | 0.344 | 0.182 | 0.307 | 0.177 | 0.299 | **0.176** | 0.299 | 0.226 | 0.344 | 0.271 | 0.315 | 0.176 | 0.315 |
| Exchange 336 | 0.347 | 0.424 | 0.350 | 0.427 | 0.377 | 0.451 | 0.402 | 0.461 | 0.331 | **0.397** | **0.301** | **0.397** | 0.367 | 0.448 | 0.460 | 0.427 | 0.313 | 0.427 |
| Exchange 720 | **0.795** | **0.668** | 0.935 | 0.732 | 1.018 | 0.771 | 1.055 | 0.767 | 0.847 | 0.691 | 0.901 | 0.714 | 0.964 | 0.746 | 1.195 | 0.695 | 0.839 | 0.695 |

(2023). Meanwhile, we also select several recent competitive models, including iTransformer Liu et al. (2023a), PatchTST Nie et al. (2023), DLinear Zeng et al. (2023), TimesNet Wu et al. (2023), and FEDformer Zhou et al. (2022b). We use ADAM as the default optimizer and report the mean squared error (MSE) and mean absolute error (MAE) as the evaluation metrics. Specifically, we use GPT2 Radford et al. (2019) as the default backbone to validate the effectiveness of Logo-LLM. We use ADAM as the default optimizer and report the mean squared error (MSE) and mean absolute error (MAE) as the evaluation metrics. For models whose default input length is not 96, we modify only the input length to 96 while keeping other settings unchanged. To ensure a fair comparison, we adopt the official implementations of the backbone models and follow their default configurations.

**Long-term Forecasting.** We conduct evaluations on 9 benchmark datasets to assess long-term forecasting. As shown in Table 1, Logo-LLM achieves better performance than most baselines. Specifically, compared to the recent SOTA methods CALF and Time-LLM, Logo-LLM yields a relative MSE reduction of 1.3% and 8.9%, respectively.

**Few-shot Forecasting.** With extensive world knowledge, LLMs exhibit excellent few-shot and zero-shot learning abilities. In NLP, prompting with a few examples can often yield comparable performance with fine-tuning. This motivates the exploration of LLMs' generalization capabilities in time series forecasting under limited supervision. Following the setup of Zhou et al. (2023), each dataset is divided into training, validation, and test sets, with only 5% training data used in few-shot scenarios. We evaluate Logo-LLM primarily on four ETT benchmarks to validate its effectiveness.

Table 2: Few-shot learning task on 5% data. The results are averaged from prediction lengths $T \in \{96, 192, 336, 720\}$.

| Methods | Logo-LLM (Ours) | | CALF (2025) | | Time-LLM (2024) | | GPT4TS (2023) | | iTransformer (2023a) | | PatchTST (2023) | | TimesNet (2023) | | FEDformer (2022b) | | DLinear (2023) | |
|---|---|---|---|---|---|---|---|---|---|---|---|---|---|---|---|---|---|---|
| Metric | MSE | MAE | MSE | MAE | MSE | MAE | MSE | MAE | MSE | MAE | MSE | MAE | MSE | MAE | MSE | MAE | MSE | MAE |
| ETTm1 | 0.540 | **0.473** | 0.611 | 0.512 | 0.648 | 0.527 | 0.626 | 0.510 | 0.731 | 0.567 | **0.526** | 0.476 | 0.717 | 0.561 | 0.730 | 0.592 | 0.572 | 0.502 |
| ETTm2 | **0.306** | **0.339** | 0.314 | 0.350 | 0.318 | 0.353 | 0.328 | 0.357 | 0.343 | 0.369 | 0.314 | 0.352 | 0.344 | 0.372 | 0.381 | 0.404 | 0.510 | 0.492 |
| ETTh1 | **0.606** | **0.519** | 0.840 | 0.613 | 0.863 | 0.640 | 0.673 | 0.557 | 0.831 | 0.624 | 0.694 | 0.569 | 0.925 | 0.647 | 0.658 | 0.562 | 0.750 | 0.611 |
| ETTh2 | **0.394** | **0.407** | 0.457 | 0.447 | 0.548 | 0.478 | 0.523 | 0.473 | 0.472 | 0.458 | 0.439 | 0.448 | 0.463 | 0.454 | 0.441 | 0.457 | 0.827 | 0.615 |

Table 3: Zero-shot learning task. The results are averaged from prediction lengths $T \in \{96, 192, 336, 720\}$. A → B indicates a zero-shot transfer setting where the model is trained on dataset A and directly evaluated on dataset B without further fine-tuning.

| Methods | Logo-LLM (Ours) | | CALF (2025) | | Time-LLM (2024) | | GPT4TS (2023) | | iTransformer (2023a) | | PatchTST (2023) | | TimesNet (2023) | | FEDformer (2022b) | | DLinear (2023) | |
|---|---|---|---|---|---|---|---|---|---|---|---|---|---|---|---|---|---|---|
| Metric | MSE | MAE | MSE | MAE | MSE | MAE | MSE | MAE | MSE | MAE | MSE | MAE | MSE | MAE | MSE | MAE | MSE | MAE |
| ETTh1 → ETTm1 | 0.762 | **0.566** | **0.755** | 0.574 | 0.847 | 0.565 | 0.798 | 0.574 | 0.825 | 0.589 | 0.894 | 0.610 | 0.794 | 0.575 | 0.765 | 0.588 | 0.760 | 0.577 |
| ETTh1 → ETTm2 | **0.313** | **0.353** | 0.316 | 0.355 | 0.315 | 0.357 | 0.317 | 0.359 | 0.320 | 0.359 | 0.318 | 0.362 | 0.339 | 0.370 | 0.357 | 0.403 | 0.399 | 0.439 |
| ETTh2 → ETTm1 | 0.841 | **0.578** | 0.836 | 0.586 | 0.868 | 0.595 | 0.920 | 0.610 | 0.912 | 0.603 | 0.871 | 0.596 | 1.286 | 0.705 | **0.741** | 0.588 | 0.778 | 0.594 |
| ETTh2 → ETTm2 | **0.316** | **0.357** | 0.319 | 0.360 | 0.322 | 0.363 | 0.331 | 0.371 | 0.329 | 0.370 | 0.420 | 0.433 | 0.361 | 0.390 | 0.365 | 0.405 | 0.496 | 0.496 |

The results of 5% few-shot learning with input length $L = 96$ are shown in Table 2. Compared with LLM-based methods Liu et al. (2025); Jin et al. (2024); Zhou et al. (2023), our proposed Logo-LLM achieves the best or second best prediction performance on all datasets. This demonstrates its excellent ability to effectively capture both local and global patterns, even under low-data settings. In comparison to CALF Liu et al. (2025) and Time-LLM Jin et al. (2024), Logo-LLM demonstrates a relative average MSE reduction of 13.5% and 19.2% respectively. These results highlight the strong data efficiency and robust generalization of Logo-LLM in low-resource forecasting scenarios.

**Zero-shot Forecasting.** We further investigate the zero-shot generalization ability of LLMs from Liu et al. (2025), where models are evaluated on a completely unseen dataset after being trained on a different source dataset, without any fine-tuning. This setup highlights the model's capacity for domain transfer and reasoning without task-specific supervision. As shown in Table 3, Logo-LLM achieves superior performance in almost all scenarios. These results demonstrate cross-domain generalization capability of our approach, underscoring its effectiveness in zero-shot learning.

## 5 ABLATION STUDIES

In this section, we conduct several ablation studies to analyze Logo-LLM. Specially, the experiments cover model fine-tuning, parameter configuration, feature mixer strategy, and the strategy of local and global modeling. More ablations on input length and different LLMs can be seen in Appendix G and H.

**Fine-tuning Strategy.** To assess different fine-tuning strategies, we conducted ablation experiments using only 5% of training data across ETT datasets. As shown in Table 4, we compare full-parameter (FP), LoRA (Hu et al., 2022), frozen fine-tuning of layer normalization and positional embedding layer (LN+PE), and the baseline, without fine-tuning (w/o FT). Among the strategies, LN+PE con-

Table 4: Ablations of 5% training data on fine-tune strategy. We select several methods to fine-tune the pre-trained LLM. The average results for all prediction lengths are listed here.

| Methods | LN+PE | | LN | | PE | | FP | | LoRA | | w/o FT | |
|---|---|---|---|---|---|---|---|---|---|---|---|---|
| Metric | MSE | MAE | MSE | MAE | MSE | MAE | MSE | MAE | MSE | MAE | MSE | MAE |
| ETTm1 | **0.540** | 0.473 | 0.543 | 0.474 | 0.541 | 0.474 | 0.537 | **0.472** | 0.549 | 0.478 | 0.541 | 0.474 |
| ETTm2 | **0.306** | **0.339** | 0.308 | 0.341 | 0.308 | 0.341 | 0.330 | 0.351 | 0.308 | 0.341 | 0.308 | 0.341 |
| ETTh1 | **0.606** | **0.519** | 0.640 | 0.542 | 0.640 | 0.542 | 0.642 | 0.543 | 0.640 | 0.543 | 0.640 | 0.542 |
| ETTh2 | **0.394** | **0.407** | 0.399 | **0.407** | 0.399 | **0.407** | 0.403 | 0.410 | 0.398 | **0.407** | 0.399 | **0.407** |

Table 5: Ablations of 5% training data on local and global features. We select different methods to align the temporal data with local and global features, respectively. The average results of all prediction lengths are listed here.

| Methods | Mixer (Ours) | | Add | | Cross | | w/o Mixer | |
|---|---|---|---|---|---|---|---|---|
| Metric | MSE | MAE | MSE | MAE | MSE | MAE | MSE | MAE |
| ETTm1 | **0.540** | **0.473** | 0.557 | **0.473** | 0.555 | 0.492 | 0.593 | 0.500 |
| ETTm2 | **0.306** | **0.339** | 0.333 | 0.360 | 0.314 | 0.352 | 0.316 | 0.350 |
| ETTh1 | **0.606** | **0.519** | 0.685 | 0.560 | 0.652 | 0.551 | 0.648 | 0.551 |
| ETTh2 | **0.394** | **0.407** | 0.564 | 0.484 | 0.496 | 0.451 | 0.593 | 0.495 |

Table 6: Ablations of 5% training data on modeling strategy. The average results of all prediction lengths are listed here. w/o represents the removal of the corresponding representation branch, and * denotes using the averaged representations from the first half of LLM layers as local features, and those from the second half as global features.

| Methods | Logo-LLM | | w/o Global | | w/o Local | | Logo-LLM* | | w/o Global* | | w/o Local* | |
|---|---|---|---|---|---|---|---|---|---|---|---|---|
| Metric | MSE | MAE | MSE | MAE | MSE | MAE | MSE | MAE | MSE | MAE | MSE | MAE |
| ETTm1 | 0.540 | 0.473 | 0.586 | 0.486 | 0.611 | 0.497 | 0.573 | 0.478 | **0.519** | **0.459** | 0.552 | 0.467 |
| ETTm2 | **0.306** | **0.339** | 0.343 | 0.364 | 0.335 | 0.360 | 0.310 | 0.344 | 0.330 | 0.354 | 0.311 | 0.343 |
| ETTh1 | **0.606** | **0.519** | 0.642 | 0.545 | 0.644 | 0.549 | 0.635 | 0.539 | 0.707 | 0.575 | 0.959 | 0.649 |
| ETTh2 | **0.394** | **0.407** | 0.414 | 0.414 | 0.403 | 0.414 | 0.586 | 0.493 | 0.612 | 0.619 | 0.604 | 0.499 |

sistently achieves superior or comparable performance across all datasets and metrics. In contrast, FP fails to show a clear advantage and even slightly underperforms on ETTm2 and ETTh2, possibly due to overfitting in small-scale datasets.

**Feature Mixer Strategy.** Table 5 presents ablation results on 5% training data to evaluate different strategies for aligning local and global features with temporal data. Specifically, we compare our proposed Mixer modules against addition (Add), cross-attention mechanisms (Cross), and a baseline without any fusion (w/o Mixer). The results show that Mixer can consistently achieve superior performance across all datasets after aligning the local and global variations, respectively, highlighting its effectiveness in capturing fine-grained interactions between temporal inputs and hierarchical feature representations from the pre-trained LLM.

**Local and Global Modeling.** Table 6 presents the ablation results using 5% training data to analyze local and global feature modeling strategies. The default Logo-LLM leverages the first and last LLM layers for local and global representations, respectively. We compare this with a variant (denoted by *) that uses the average of the first and second half from all LLM layers as local and global features, respectively. The results demonstrate that selecting the first and last layers yields better performance, indicating that boundary-layer features capture more distinct local and global variations than uniformly averaged intermediate representations.

**The Number of LLM Layers.** To leverage the representational ability of the pre-trained LLM while controlling computational overhead, we select a subset of layers to balance performance and efficiency. Therefore, we conduct ablations on numbers of LLM layers on ETTm1 and ETTm2 using 5% training data. Additionally, we choose the LLM-based method CALF (Liu et al., 2025) as a comparison.

As shown in Figure 3, just two Transformer layers are sufficient to unlock most of the LLM's representational power for time series modeling. As the depth increases, the predictive performance improves and reaches its peak around 6 layers. This highlights that early layers in the pre-trained LLM already encode rich local and global temporal patterns. Moreover, there exists a sweet spot in model depth where the trade-off between long-term dependencies and fine-grained variations is balanced. This observation validates our design of selectively leveraging shallow and deep layer representations, rather than relying on the last layer.

**Impact of Local Feature Layer Selection.** To investigate the optimal layer for extracting local representations from the pre-trained LLM, we conducted an ablation study by varying the selected Transformer layer to represent local features. As shown in Figure 4, we evaluated the performance of Logo-LLM with MSE as the evaluation criterion.

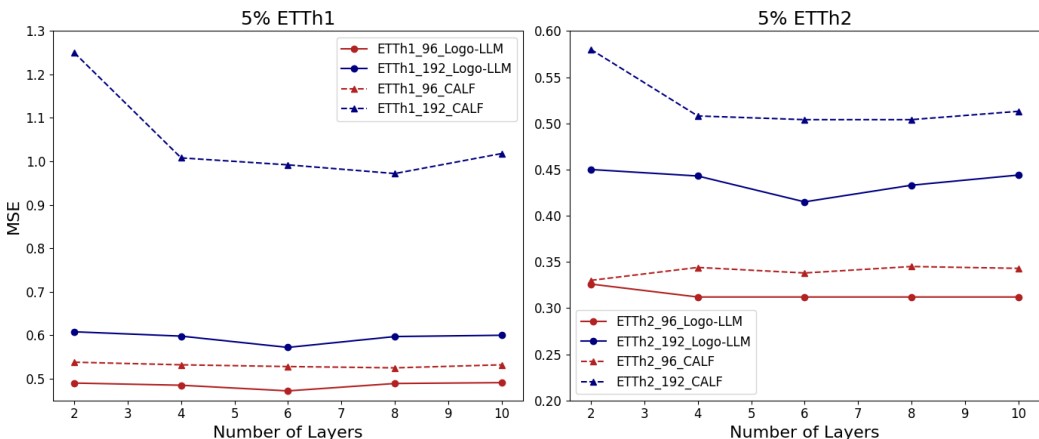

Figure 3: Comparison of Logo-LLM and CALF with various numbers of layers on ETTh1 and ETTh2 datasets. The prediction length is set as {96, 192} with input length $L = 96$.

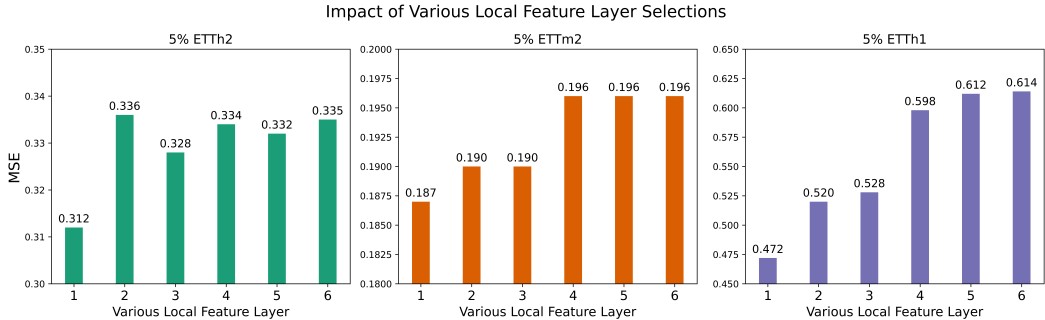

Figure 4: Visualization of different selections {1, 2, 3, 4, 5, 6} about local feature layer on ETTh1, ETTm2, and ETTh2. The prediction length is set as 96 with input length $L = 96$.

We observe that using the first-layer output as a local feature yields the best performance and performance gradually deteriorates or plateaus when deeper layers are used. This finding supports our design choice and aligns with the representational hierarchy of LLMs. Early Transformer layer in LLMs tend to preserve fine-grained temporal patterns that are more directly correlated with the original input, whereas deeper layers abstract away local variations in favor of global semantics. By designing local modeling at the shallowest layer, Logo-LLM can effectively capture short-term fluctuations that are critical for time series forecasting.

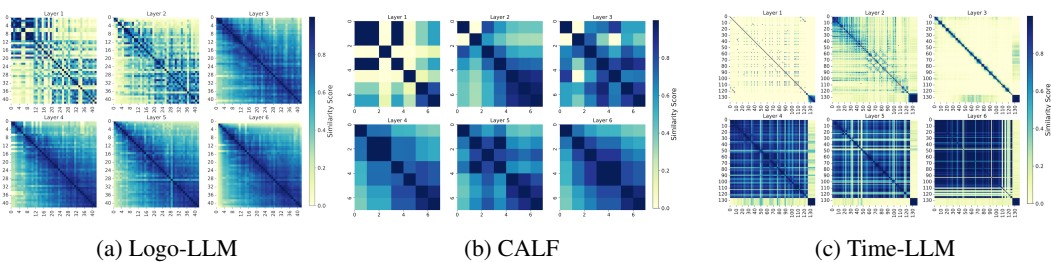

(a) Logo-LLM         (b) CALF         (c) Time-LLM

Figure 5: Similarity matrices of each patch across Transformer layers in (a) Logo-LLM (b) CALF and (c) Time-LLM, illustrating that shallow layers exhibit pronounced local patterns while deeper layers capture broader global dependencies.

Table 7: Comparison of LLM-based time series forecasting methods in terms of total parameters (Param.), training parameters (Param.$^*$), computation time for a single run and MSE. The input and prediction lengths are set as 96 and 720 respectively.

| Methods | Logo-LLM (Ours) | | | | CALF (2025) | | | | Time-LLM (2024) | | | | GPT4TS (2023) | | | |
|---|---|---|---|---|---|---|---|---|---|---|---|---|---|---|---|---|
| | Param. | Param.$^*$ | Time | MSE | Param. | Param.$^*$ | Time | MSE | Param. | Param.$^*$ | Time | MSE | Param. | Param.$^*$ | Time | MSE |
| Weather | 92.1M | 11.0M | 1.05ms | 0.350 | 180.7M | 18.5M | 10.95ms | 0.355 | 6653.1M | 45.7M | 192.3ms | 0.368 | 88.6M | 7.5M | 0.93ms | 0.360 |
| ETTh1 | 92.1M | 11.0M | 0.83ms | 0.446 | 180.7M | 18.5M | 8.99ms | 0.479 | 6653.1M | 45.7M | 210.6ms | 0.483 | 88.6M | 7.5M | 0.76ms | 0.495 |

**Local and Global Representation Analysis.** To provide empirical validation for our hypothesis that shallow and deep layers of LLM specialize in the extraction of local and global patterns, we visualize the cosine similarity matrices of hidden representations across all six layers in Figure 5. In the early layers (Layers 1–2), we observe that similarity scores are highly concentrated around the diagonal. This indicates that each patch predominantly attends to its temporal neighbors, confirming that shallow layers focus on short-term patterns and local consistency. In contrast, some rows in the similarity matrix exhibit local clustering patterns, where high similarity scores appear in off-diagonal regions. This behavior is reminiscent of how LLMs process textual data, capturing syntactic or surface-level relationships in earlier layers and progressively learning semantic or abstract representations in deeper layers (Lee et al., 2024; Liu et al., 2024). This suggests that the model has captured local structures, even when the corresponding patches are not temporally adjacent in the sequence.

As the model progresses to deeper layers (Layers 3–6), the similarity matrices exhibit increasingly smooth and uniformly distributed patterns. Elevated similarity is observed even between distant patches, indicating a transition from localized representations toward more global and abstract ones. At this stage, the model begins to integrate information across the entire sequence, allowing it to capture long-range dependencies, overarching trends structures. This layer-wise transition, from local pattern extraction in the shallow layers to global temporal abstraction in the deeper layers, is well aligned with the architectural design of Logo-LLM. In particular, we employ a Local Mixer on early-layer outputs to effectively capture short-term local variations, and a Global Mixer on the deeper-layer representations to model long-range dependencies. This hierarchical decomposition enables Logo-LLM to learn temporal dynamics at multiple scales, enhancing its ability to respond to short-term fluctuations while maintaining long-range forecasting capabilities.

**Time Cost Analysis.** We conducted experiments about time cost on two datasets: Weather and ETTh1, and the input and prediction lengths are set to 96 and 720, respectively. The batch size is set to 1 and we test all models on same GPU. As shown in Table 7, Logo-LLM shows significant improvements in both efficiency and accuracy compared to CALF (Liu et al., 2025) and Time-LLM (Jin et al., 2024). Moreover, Logo-LLM performs more fine-grained modeling of local and global temporal features than GPT4TS (Zhou et al., 2023). Although this leads to a marginal increase in inference latency, it results in a greater improvement in predictive performance.

## 6 CONCLUSION

In this paper, we propose Logo-LLM, a novel framework that utilizes the hierarchical feature extraction of pre-trained large language models (LLMs) for time series forecasting. Unlike previous approaches, Logo-LLM leverages both shallow and deep layers of LLMs to capture local and global temporal features separately. The Local-Mixer and Global-Mixer modules align these features with time series data, enhancing the model's ability to capture fine-grained fluctuations and long-range dependencies. The experiments across real-world datasets demonstrate that Logo-LLM outperforms existing methods in long-term, few-shot, and zero-shot learning tasks, offering superior generalization and low computational overhead. This work highlights the potential of LLMs in time series modeling and opens avenues for further improvements in forecasting accuracy and efficiency.

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

## A  THE USE OF LARGE LANGUAGE MODELS (LLMS)

The central technical contribution of this paper relies on a pre-trained LLM as the backbone for hierarchical semantic feature extraction, which is an essential component of the proposed Logo-LLM framework. We adopted the pre-trained GPT-2 model (Radford et al., 2019) as the core encoder. Its hierarchical Transformer architecture enables the capture of multi-scale temporal patterns, which aligns with our research goal of decoupling local and global modeling for time series forecasting. Furthermore, we used a commercial LLM for limited, auxiliary manuscript polishing, with strict human oversight to ensure accuracy, consistency, and scientific integrity.

## B  MOTIVATION

Despite the growing interest in adapting Large Language Models (LLMs) for time series forecasting (Liu et al., 2025; Jin et al., 2024; Zhou et al., 2023), existing approaches are constrained by a fundamental limitation: they predominantly treat the LLM as a black-box encoder. As illustrated in Figure 1, these methods typically leverage only the final-layer output of the LLM for prediction. While this practice inherits the strong global dependency modeling capacity of the Transformer architecture, it overlooks the rich, hierarchical representations inherent to the LLM, which are distributed across its different internal layers.

Research in computer vision and natural language processing has demonstrated that Transformer architectures learn a hierarchical representational structure: shallow layers tend to capture local, fine-grained patterns, while deeper layers encode more global, semantic information. Through systematic empirical analysis (see Figure 5), we confirm that this progressive abstraction from local to global patterns is also present when LLMs process time series data. Specifically, we observe that the outputs of shallow LLM layers exhibit a higher sensitivity to short-term local fluctuations in time series, whereas the outputs of deeper layers are more adept at capturing long-range global dependencies.

The novelty of our work does not stem from merely applying the concept of "local and global modeling" to time series, as this idea has already been explored in traditional time series models. Instead, our core conceptual breakthrough lies in being the first to systematically reveal and leverage the inherent hierarchical representational properties of pre-trained LLMs, which exist even without fine-tuning, to naturally address multi-scale modeling challenges in time series data.

Unlike previous models that require specially designed local and global modules (Dai et al., 2024; Chen et al., 2024), our work points to a new direction: the internal structure of a pre-trained LLM itself serves as an off-the-shelf, powerful multi-scale feature extractor. Our contribution lies in the discovery of this property in the context of time series and the creative utilization of this insight to propose a novel modeling paradigm. Specifically, our novelty is reflected in the following aspects:

- We advance the paradigm of using LLMs for time series from black-box encoders to white-box feature mining. Rather than treating the LLM as a monolithic encoder, we regard it as a rich repository of hierarchical features, actively extracting semantically distinct representations from different layers.

- Unlike methods that require pre-training hierarchical models from scratch on time series data, Logo-LLM directly utilizes readily available, generically pre-trained GPT-2. This demonstrates that the internal representations of general-purpose LLMs inherently possess sufficient transferability to time series, which is a significant finding in its own right.

- Building on the above insight, we design two lightweight modules: the Local-Mixer and Global-Mixer. The innovation of these modules lies in their specialized design; they are not generic fusion layers but are explicitly tailored to handle the heterogeneous features from different LLM layers (local details from shallow layers and global semantics from deep layers), enabling more refined feature alignment and information fusion.

## C  DETAILED EXPERIMENTAL SETTINGS

We use ADAM as the default optimizer and report the mean squared error (MSE) and mean absolute error (MAE) as the evaluation metrics. A lower MSE/MAE indicates a better performance. All experiments are implemented using PyTorch and conducted with a fixed random seed on two NVIDIA H20 GPUs (96GB each). For models (Zhou et al., 2023; Jin et al., 2024; Nie et al., 2023; Zeng et al., 2023) whose default input length is not 96, we modify only the input length to 96 while keeping other settings unchanged. To ensure a fair comparison, we adopt the official implementations of the backbone models and follow their default configurations. Most of the baseline results are taken from the original papers of CALF (Liu et al., 2025) and iTransformer (Liu et al., 2023a).

Table 8: The statistical details of benchmark datasets.

| Dataset | Size | Frequency | Length | Prediction Length |
|---------|------|-----------|--------|-------------------|
| ETTm1&m2 | 7 | 15min | 69680 | {96, 192, 336, 720} |
| ETTh1&h2 | 7 | 1hour | 17420 | {96, 192, 336, 720} |
| Weather | 21 | 10min | 52696 | {96, 192, 336, 720} |
| ECL | 321 | 1hour | 26304 | {96, 192, 336, 720} |
| Traffic | 862 | 1hour | 17544 | {96, 192, 336, 720} |
| ILI | 7 | 7days | 966 | {24, 36, 48, 60} |
| Exchange | 8 | 1day | 7588 | {96, 192, 336, 720} |

## D  DATASETS

**ETT**Zhou et al. (2021)  contains two sub-datasets: ETT1 and ETT2, collected from two electricity transformers at two stations. Each of them has two versions in different resolutions (15min & 1h). The ETT dataset contains multiple series of loads and one series of oil temperatures.

**Electricity Consuming Load (ECL)**[1]  corresponds to the electricity consumption (Kwh) of 321 clients.

**Weather**[2]  contains 21 meteorological indicators, such as air temperature, humidity, etc, recorded every 10 minutes for the entirety of 2020.

**Exchange**  collects the daily exchange rates of 8 countries (Australia, British, Canada, Switzerland, China, Japan, New Zealand, and Singapore) from 1990 to 2016.

**National Illness (ILI)**[3]  corresponds to the weekly recorded influenza-like illness patients from the US Center for Disease Control and Prevention.

The datasets are characterized as follows in Table 8. We follow standard protocol Nie et al. (2023) and split all datasets into training, validation, and test sets in chronological order by the ratio of 6:2:2 for the ETT dataset and 7:1:2 for the other datasets. Frequency indicates the sampling time difference between neighboring time steps.

## E  BASELINE METHODS

Our baseline methods are described as follows:

- CALF is a novel framework that enhances multivariate time series forecasting by aligning the distribution discrepancy between textual and temporal inputs through cross-modal matching, feature regularization, and output consistency. The code is available at https://github.com/Hank0626/CALF.

- Time-LLM is a reprogramming framework that reprogrammes frozen large language models for general time series forecasting by aligning time series and language modalities through text-based input transformation and a novel Prompt-as-Prefix (PaP) mechanism. The official code is available at https://github.com/KimMeen/Time-LLM.

- GPT4TS is a framework that leverages language models pre-trained on large-scale data for general time series analysis, achieving strong performance across diverse tasks without modifying the core architecture of the original transformer. The code is available at https://github.com/DAMO-DI-ML/NeurIPS2023-One-Fits-All.

- iTransformer is a Transformer-based approach which innovatively introduces an inverted perspective to model time-series data for excellent performance and versatility. The code is available at https://github.com/thuml/iTransformer.

- PatchTST is a Transformer-based method by patching and channel independence strategies for multivariate time series forecasting. The official code is available at https://github.com/yuqinie98/patchtst.

---

[1]https://archive.ics.uci.edu/ml/datasets/ElectricityLoadDiagrams20112014

[2]https://www.bgc-jena.mpg.de/wetter/

[3]https://gis.cdc.gov/grasp/fluview/fluportaldashboard.html

- TimesNet is a CNN-based method which transforms 1D sequence into 2D tensor to capture the periodicity of the time series. The code is available at `https://github.com/thuml/TimesNet`.

- FEDformer is a forecasting model that integrates seasonal-trend decomposition and frequency-domain enhancement to effectively and efficiently capture both global trends and detailed patterns in time series, achieving superior accuracy with linear computational complexity. The code is available at `https://github.com/MAZiqing/FEDformer`.

- DLinear is an MLP-based approach which challenges the Transformer-based methods with a one-layer linear layer. The source code is available at `https://github.com/cure-lab/LTSF-Linear`.

## F LOCAL AND GLOBAL REPRESENTATIONS

Our motivation is inspired by the hierarchical representation structure observed in pre-trained large language models. We begin by visualizing the similarity matrices of hidden representations across different layers in our proposed Logo-LLM, as illustrated in Figure 5. To systematically analyze this phenomenon, we further visualize layer-wise similarity matrices for two representative baselines, CALF (Liu et al., 2025) and Time-LLM (Jin et al., 2024), with the number of layers standardized to 6. As illustrated in Figure 5b and 5c, the visualizations reveal a consistent hierarchical pattern across all models: shallow layers tend to focus on local variations and fine-grained patterns, while deeper layers progressively capture global trends and long-range dependencies. These findings suggest that the pre-trained LLM inherently develops hierarchical semantics when applied to time series forecasting, enabling the LLM to extract multi-scale temporal information effectively.

## G ABLATION ON INPUT LENGTH

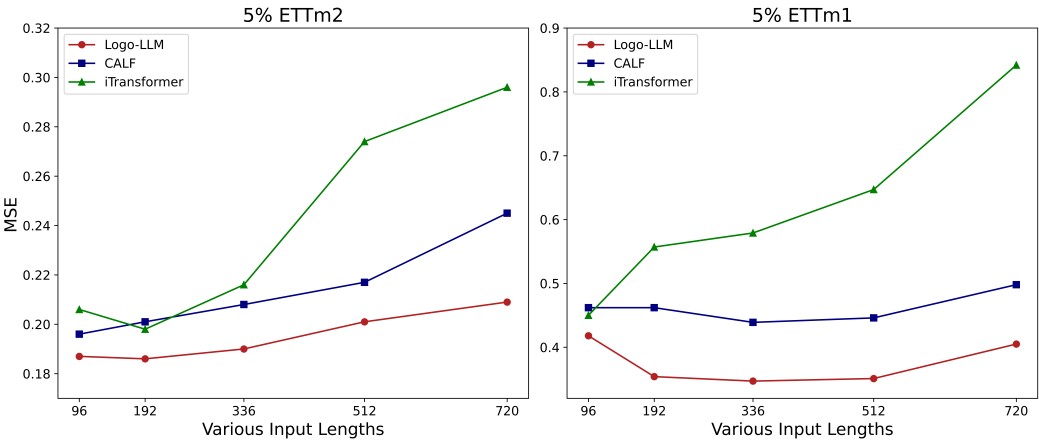

Figure 6: Ablation study on various input lengths. We conducted experiments with the input length $L = \{96, 192, 336, 512, 720\}$ and the prediction length $T = 96$ on 5% training data of ETTm1 and ETTm2 datasets. The figures demonstrate the strong adaptability and robustness of our model when processing temporal contexts of varying lengths.

Figure 6 presents the performance comparison of Logo-LLM, CALF Liu et al. (2025), and iTransformer Liu et al. (2023a) across different input lengths on ETTm1 and ETTm2 datasets using MSE as the evaluation metric. From the figures, Logo-LLM consistently achieves the lowest MSE under all input length settings. This highlights the benefit of our layer-wise local-global modeling strategy, better exploiting LLM's semantic hierarchy and filtering useful temporal cues from long contexts. For all models, increasing the input length does not always lead to better performance, indicating that simple extension of input horizon may introduce more noise or redundant dependencies in few-shot learning scenarios.

Table 9: Full results of Few-shot learning task on 5% data with the input length $L = 96$. The prediction lengths $T$ are set as $\{96, 192, 336, 720\}$. '-' means that 5% time series is not sufficient to constitute a training set.

| Methods | | Logo-LLM (Ours) | | CALF (2025) | | Time-LLM (2024) | | GPT4TS (2023) | | iTransformer (2023a) | | PatchTST (2023) | | TimesNet (2023) | | FEDformer (2022b) | | DLinear (2023) | |
|---|---|---|---|---|---|---|---|---|---|---|---|---|---|---|---|---|---|---|---|
| Metric | | MSE | MAE | MSE | MAE | MSE | MAE | MSE | MAE | MSE | MAE | MSE | MAE | MSE | MAE | MSE | MAE | MSE | MAE |
| ETTm1 | 96 | 0.418 | 0.406 | 0.462 | 0.438 | 0.560 | 0.477 | 0.545 | 0.472 | 0.450 | 0.473 | 0.399 | 0.414 | 0.606 | 0.518 | 0.628 | 0.544 | 0.437 | 0.430 |
| | 192 | 0.517 | 0.446 | 0.516 | 0.468 | 0.574 | 0.495 | 0.548 | 0.473 | 0.592 | 0.508 | 0.441 | 0.436 | 0.681 | 0.539 | 0.666 | 0.566 | 0.496 | 0.462 |
| | 336 | 0.575 | 0.474 | 0.618 | 0.525 | 0.596 | 0.504 | 0.619 | 0.503 | 0.730 | 0.568 | 0.499 | 0.467 | 0.786 | 0.597 | 0.807 | 0.628 | 0.562 | 0.503 |
| | 720 | 0.691 | 0.565 | 0.849 | 0.618 | 0.861 | 0.632 | 0.790 | 0.592 | 1.151 | 0.719 | 0.767 | 0.587 | 0.796 | 0.593 | 0.822 | 0.633 | 0.792 | 0.612 |
| | Avg | 0.540 | 0.473 | 0.611 | 0.512 | 0.648 | 0.527 | 0.626 | 0.510 | 0.731 | 0.567 | 0.526 | 0.476 | 0.717 | 0.561 | 0.730 | 0.592 | 0.572 | 0.502 |
| ETTm2 | 96 | 0.187 | 0.266 | 0.196 | 0.278 | 0.200 | 0.282 | 0.196 | 0.280 | 0.206 | 0.292 | 0.206 | 0.288 | 0.220 | 0.299 | 0.229 | 0.320 | 0.305 | 0.386 |
| | 192 | 0.255 | 0.310 | 0.269 | 0.326 | 0.267 | 0.325 | 0.267 | 0.324 | 0.284 | 0.343 | 0.264 | 0.324 | 0.311 | 0.361 | 0.394 | 0.361 | 0.413 | 0.450 |
| | 336 | 0.319 | 0.348 | 0.338 | 0.364 | 0.328 | 0.360 | 0.323 | 0.353 | 0.338 | 0.363 | 0.334 | 0.367 | 0.338 | 0.427 | 0.378 | 0.427 | 0.482 | 0.487 |
| | 720 | 0.465 | 0.434 | 0.454 | 0.431 | 0.478 | 0.445 | 0.525 | 0.476 | 0.543 | 0.476 | 0.454 | 0.432 | 0.509 | 0.510 | 0.523 | 0.510 | 0.839 | 0.646 |
| | Avg | 0.306 | 0.339 | 0.314 | 0.350 | 0.318 | 0.353 | 0.328 | 0.357 | 0.343 | 0.369 | 0.314 | 0.352 | 0.344 | 0.372 | 0.381 | 0.404 | 0.510 | 0.492 |
| ETTh1 | 96 | 0.472 | 0.448 | 0.528 | 0.490 | 0.576 | 0.519 | 0.513 | 0.478 | 0.632 | 0.538 | 0.557 | 0.519 | 0.892 | 0.625 | 0.593 | 0.529 | 0.547 | 0.503 |
| | 192 | 0.572 | 0.508 | 0.992 | 0.670 | 0.818 | 0.635 | 0.732 | 0.592 | 0.917 | 0.669 | 0.711 | 0.570 | 0.940 | 0.665 | 0.652 | 0.563 | 0.720 | 0.604 |
| | 336 | 0.737 | 0.587 | 1.000 | 0.680 | 1.194 | 0.765 | 0.774 | 0.602 | 0.944 | 0.665 | 0.816 | 0.619 | 0.945 | 0.653 | 0.731 | 0.594 | 0.984 | 0.727 |
| | 720 | - | - | - | - | - | - | - | - | - | - | - | - | - | - | - | - | - | - |
| | Avg | 0.606 | 0.519 | 0.840 | 0.613 | 0.863 | 0.640 | 0.673 | 0.557 | 0.831 | 0.624 | 0.694 | 0.569 | 0.925 | 0.647 | 0.658 | 0.562 | 0.750 | 0.611 |
| ETTh2 | 96 | 0.312 | 0.349 | 0.338 | 0.372 | 0.325 | 0.363 | 0.330 | 0.361 | 0.415 | 0.426 | 0.401 | 0.421 | 0.409 | 0.420 | 0.390 | 0.424 | 0.442 | 0.456 |
| | 192 | 0.415 | 0.418 | 0.504 | 0.473 | 0.419 | 0.420 | 0.417 | 0.420 | 0.489 | 0.464 | 0.452 | 0.455 | 0.483 | 0.464 | 0.457 | 0.465 | 0.617 | 0.542 |
| | 336 | 0.456 | 0.453 | 0.528 | 0.495 | 0.899 | 0.652 | 0.823 | 0.639 | 0.511 | 0.485 | 0.464 | 0.469 | 0.499 | 0.479 | 0.477 | 0.483 | 1.424 | 0.849 |
| | 720 | - | - | - | - | - | - | - | - | - | - | - | - | - | - | - | - | - | - |
| | Avg | 0.394 | 0.407 | 0.457 | 0.447 | 0.548 | 0.478 | 0.523 | 0.473 | 0.472 | 0.458 | 0.439 | 0.448 | 0.463 | 0.454 | 0.441 | 0.457 | 0.827 | 0.615 |

Table 10: Ablations of 100% data on different LLMs with the input length $L = 96$. The prediction lengths $T$ are set as $\{96, 192, 336, 720\}$.

| Methods | | GPT-2 | | BERT | |
|---|---|---|---|---|---|
| Metric | | MSE | MAE | MSE | MAE |
| ETTm1 | 96 | 0.317 | 0.343 | 0.316 | 0.343 |
| | 192 | 0.368 | 0.370 | 0.368 | 0.372 |
| | 336 | 0.394 | 0.393 | 0.392 | 0.394 |
| | 720 | 0.460 | 0.430 | 0.453 | 0.430 |
| ETTm2 | 96 | 0.175 | 0.253 | 0.174 | 0.251 |
| | 192 | 0.243 | 0.298 | 0.241 | 0.296 |
| | 336 | 0.304 | 0.338 | 0.302 | 0.336 |
| | 720 | 0.408 | 0.400 | 0.402 | 0.394 |
| ETTh1 | 96 | 0.373 | 0.388 | 0.372 | 0.391 |
| | 192 | 0.413 | 0.412 | 0.419 | 0.418 |
| | 336 | 0.434 | 0.424 | 0.454 | 0.439 |
| | 720 | 0.446 | 0.447 | 0.457 | 0.456 |
| ETTh2 | 96 | 0.274 | 0.330 | 0.298 | 0.347 |
| | 192 | 0.344 | 0.376 | 0.361 | 0.387 |
| | 336 | 0.369 | 0.398 | 0.374 | 0.399 |
| | 720 | 0.394 | 0.417 | 0.392 | 0.423 |

## H  ABLATION ON LARGE LANGUAGE MODELS

To further validate our findings, we extended our experiments to apply LLMs with diverse architectures, i.e., the encoder-based BERT (Devlin et al., 2019). As shown in Table 10, the performance gap between the two models across multiple ETT datasets is negligible. This indicates that hierarchical specialization capabilities, where shallow layers capture local patterns and deeper layers encode global dependencies, are not unique to GPT-2 (Radford et al., 2019). Instead, this capability exists as a universal intrinsic property of LLMs, independent of specific architectural designs.

## I  FULL RESULTS OF FEW-SHOT LEARNING

Following the findings of Zeng et al. (2023) and Nie et al. (2023), which demonstrate the effectiveness of the channel-independence strategy for time series data, we treat each multivariate time series

as a collection of independent univariate series. Consistent with standard experimental protocols Liu et al. (2025), each time series is divided into three subsets: training, validation, and testing. For the few-shot forecasting setting, only 5% of the training timesteps is used, while the validation and test sets remain unchanged. The evaluation metrics are the same as those used in conventional multivariate time series forecasting. As shown in Table 9, each experiment is repeated three times, and we report the average results.

## J    LIMITATIONS

While Logo-LLM demonstrates promising performance in time series forecasting, several limitations remain. First, the current method primarily relies on empirical observations to guide the selection of shallow and deep layers for local and global modeling, lacking a more rigorous theoretical analysis of the layer-wise representations. Then, Logo-LLM is evaluated mainly on standard benchmarks and its effectiveness under real-world distribution shifts or domain-specific anomalies remains to be explored. Finally, Logo-LLM mainly leverages the local and global temporal patterns from the hierarchical features of LLMs, yet the rich world knowledge embedded in pre-trained LLMs remains underutilized. How to fully exploit this knowledge and design more powerful frameworks that seamlessly integrate LLMs for time series forecasting warrants further investigation.

