# OpenReview forum: "Local and Global Modeling with Large Language Models for Time Series Forecasting"
_ICLR.cc/2026/Conference — ICLR 2026 Conference Withdrawn Submission_

### Official Review · Reviewer_tUMa · 2025-10-25

**Soundness:** 2
**Presentation:** 3
**Contribution:** 2
**Rating:** 2
**Confidence:** 5

**Summary:**

The paper introduces Logo-LLM, a framework for time series forecasting that explicitly leverages hierarchical layer representations within pre-trained LLMs. By decoupling local and global modeling—using early layers for short-term dynamics and deeper layers for long-range dependencies—the method improves interpretability and efficiency. The proposed Local-Mixer and Global-Mixer modules align these multi-scale features with temporal inputs, achieving superior long-term, few-shot, and zero-shot forecasting results across multiple benchmarks, while maintaining low computational overhead.

**Strengths:**

1. Demonstrates that LLM layers naturally encode different temporal scales, turning a black-box encoder into a hierarchical feature extractor.
2. Freezes most LLM parameters while adding simple Mixer modules, achieving strong performance with minimal computation.
3. Covers long-term, few-shot, and zero-shot settings across diverse benchmarks, with clear performance gains.

**Weaknesses:**

1. The idea mainly extends known “multi-scale” intuitions to LLM layers, not a fundamentally new forecasting paradigm, and improvements (1–3%) over strong baselines are modest.

2. The use of LLMs is not sufficiently motivated, as the method lacks textual information to assist time series modeling. It may be worth exploring benchmarks or datasets that combine context with time series data [1] [2] [3].

3. Certain related works or baselines, such as TEMPO [4], are missing.

4. The theoretical justification for why shallow/deep LLM layers correspond to local/global features is observational, not analytical.

[1] Context is Key: A Benchmark for Forecasting with Essential Textual Information

[2] Time-MMD: Multi-Domain Multimodal Dataset for Time Series Analysis

[3] From News to Forecast: Integrating Event Analysis in LLM-Based Time Series Forecasting with Reflection

[4]  TEMPO: Prompt-based Generative Pre-trained Transformer for Time Series Forecasting.

**Questions:**

Have you considered using text related to time series to assist with time series forecasting? and some new benchmarks? like [1] [2]

[1] Context is Key: A Benchmark for Forecasting with Essential Textual Information

[2] MTBench: A Multimodal Time Series Benchmark for Temporal Reasoning and Question Answering

---

### Official Review · Reviewer_S64E · 2025-10-28

**Soundness:** 2
**Presentation:** 3
**Contribution:** 1
**Rating:** 0
**Confidence:** 5

**Summary:**

This paper introduces Logo-LLM, a forecasting model that leverages shallow and deep layers of a frozen GPT-2 to capture local and global time-series patterns. Two lightweight mixers align these features for prediction, achieving strong results across benchmarks with limited tuning. While effective, the method’s novelty is minimal, as it mainly adds the mixer module to the GPT4TS framework.

**Strengths:**

- Targeted ablations justify the first/last-layer design. First+last beats averaging halves across ETT, local-layer sweep shows the first (shallow) layer works best, and layer-count ablation shows small stacks (≈2–6) capture most gains.

**Weaknesses:**

- Novelty is very limited. Compared to GPT4TS the main incremental change is adding the Mixer modules, while other pieces like patching, instance normalization, and a largely frozen LLM backbone are similar.
- Lacks a substantive discussion of existing methods for jointly modeling global and local dynamics in time series, for example multi-scale architectures with pyramidal or recursive downsampling and using different downsampling rates.
- Missing citation and empirical comparison against contemporaneous multi-scale approaches, for example LLM-Mixer: Multiscale Mixing in LLMs for Time Series Forecasting and Adaptive Multi-Scale Decomposition Framework, which weakens both novelty and evaluation completeness.

**Questions:**

Please address the identified weaknesses and limitations noted above.

---

### Official Review · Reviewer_YAA5 · 2025-10-31

**Soundness:** 3
**Presentation:** 3
**Contribution:** 2
**Rating:** 2
**Confidence:** 5

**Summary:**

The paper proposes Logo-LLM, a LLM-based framework for time series forecasting. The authors hypothesize that shallow LLM layers capture fine-grained local patterns while deeper layers encode long-range dependencies. Based on the hypothesis, Logo-LLM extracts shallow and deep layer features from a frozen/few-tuned LLM and interprets them as local vs global temporal features, then it aligns those features to time-series patches via two lightweight MLP Mixer modules (Local-Mixer and Global-Mixer). The evaluation across long-term forecasting, few-shot, and zero-shot settings shows the generalization capability of Logo-LLM with superior performances against baselines while maintaining low computational overhead.

**Strengths:**

1. The idea of leveraging different LLM layers capture different temporal granularities is intuitive and well-motivated.
2. Extracting shallow and deep LLM representations and fusing them with two lightweight MLP Mixers is straightforward to implement and computationally efficient compared to full fine-tuning approaches.
3. The paper is well-organized and easy to follow, with clear figures and explanations of each module.

**Weaknesses:**

1. Limited Technical Novelty. The method mainly combines existing concepts of layer-wise feature extraction and light-weight fusion, which has been extensively explored in vision [1], audio [2] and time series [3] domains. The contribution of replacing the underlying architecture to a pre-trained LLM is primarily engineering not scientific.

2. Experiment Rigor: Although the few-shot and zero-shot results are intriguing, the evaluation is based on a limited number of trials (only three runs) and uses a single LLM backbone (GPT-2), which limits the statistical reliability and generality of the conclusions.




[1] Sun, Huixin, et al. "MAFormer: A transformer network with multi-scale attention fusion for visual recognition." Neurocomputing 595 (2024): 127828.
[2] Chen, Yafeng, et al. "An enhanced res2net with local and global feature fusion for speaker verification." arXiv preprint arXiv:2305.12838 (2023).
[3]Tonekaboni, Sana, et al. "Decoupling local and global representations of time series." International Conference on Artificial Intelligence and Statistics. PMLR, 2022.

**Questions:**

1. The runtime comparison uses batch = 1. Could you provide throughput (samples/sec), GPU memory, and training time per epoch for realistic batch sizes?
2. Why was the evaluation conducted with only one random seed for zero-shot results? How stable are the results across seeds?
3. Have you tested whether the hierarchical behavior (local, global) persists across other LLM architectures (e.g., LLaMa, BERT or GPT-Neo)?

---

### Official Review · Reviewer_v9t2 · 2025-11-01

**Soundness:** 1
**Presentation:** 3
**Contribution:** 1
**Rating:** 2
**Confidence:** 5

**Summary:**

This paper proposes Logo-LLM by considering local and global information from the LLM for time-series forecasting. Specifically, Logo-LLM fuses hidden states from inner LLM layers jointly with Local-Mixer and Global-Mixer modules. The experiments for time-series forecasting demonstrate effectiveness for few-show and zero-show scenarios.

**Strengths:**

1. This paper is easy to read, and the method is intuitive to understand.
2. The experiments are extensive, especially for ablation studies that examine different strategies for manipulating inner LLM modules.

**Weaknesses:**

1. It's challenging to spot the technical novelty in this paper. The proposed Logo-LLM is another work that jointly considers local and global information, although the definition of local and global may be different from existing works. However, such decoupling ideas have been explored in multiple existing works, e.g., [1, 2, 3]. The authors should also discuss more on this line of related works in Sec. 2.2.

[1] Time-LLM: Time Series Forecasting by Reprogramming Large Language Models

[2] TimeMixer: Decomposable Multiscale Mixing for Time Series Forecasting

[3] LLM4TS: Aligning Pre-Trained LLMs as Data-Efficient Time-Series Forecasters

2. Additionally, it remains unclear why using hidden states from inner LLM layers could benefit time-series forecasting. For instance, [4] uncovers that excluding LLM could even improve time-series forecasting performance. In that case, this work utilizing inner LLM layers sounds less convincing to me.

[4] Are Language Models Actually Useful for Time Series Forecasting?

3. The baselines are not state-of-the-art. The authors only include CALF and Time-LLM as the representative baselines since 2024; however, there are multiple time-series baselines published this year. For instance, please refer to the baselines in [5] (no need to compare with KAIROS, which can be considered as concurrent works).

[5] KAIROS: Unified Training for Universal Non-Autoregressive Time Series Forecasting

4. While it is appreciated to have extensive ablation studies, it sounds to me that this paper merely tested the variants of the LLM but may not be conclusive to serve as the general guidelines that can benefit the community, since Table 4 only performs with the ETT-related benchmarks.

**Questions:**

1. Why do the authors incorporate input embeddings together with global and local features? What would Logo-LLM perform without merging it after the LLM?

---

### Note · Authors · 2025-11-17

I have read and agree with the venue's withdrawal policy on behalf of myself and my co-authors.